# Amazon rainforest adjusts to long-term experimental drought

Pablo Sanchez-Martinez ⬡ [1]✉, Lion R. Martius ⬡ [1], Paulo Bittencourt[2,3], Mateus Silva[2], Oliver Binks[4], Ingrid Coughlin[5], Vanessa Negrão-Rodrigues ⬡ [6,7], João Athaydes Silva Jr[6,8], Antonio Carlos Lola Da Costa[6,8], Rachel Selman ⬡ [1], Sami Rifai ⬡ [9], Lucy Rowland ⬡ [2], Maurizio Mencuccini ⬡ [4,10] & Patrick Meir[1]

Drought-induced mortality is expected to cause substantial biomass loss in the Amazon basin. However, rainforest responses to prolonged drought are largely unknown. Here, we demonstrate that an Amazonian rainforest plot subjected to more than two decades of large-scale experimental drought reached eco-hydrological stability. After elevated tree mortality during the first 15 years, ecosystem-level structural changes resulted in the remaining trees no longer experiencing drought stress. The loss of the largest trees led to increasing water availability for the remaining trees, stabilizing biomass in the last 7 years of the experiment. Hydraulic variables linked to physiological stress, such as leaf water potential, sap flow and tissue water content, converged to the values observed in a corresponding non-droughted control forest, indicating hydraulic homeostasis. While it prevented drought-induced collapse, eco-hydrological stabilization resulted in a forest with reduced biomass and carbon accumulation in wood. These findings show how tropical rainforests may be resilient to persistent soil drought.

The Amazon rainforest is one of the largest terrestrial ecosystem carbon pools on Earth, playing an important role in global climate dynamics through the exchange of large quantities of $CO_2$ and energy with the atmosphere[1–3]. However, there is increasing evidence that the Amazon carbon sink is at risk of switching to a temporary source in some drought years[4] and may be decreasing in size over the long term because of, in part, to losses of carbon from biomass related to an increase in tree mortality rates, most probably related to warming and drying in some regions[5–9]. Tree mortality can also reduce evapotranspiration, with large additional impacts on atmospheric water recycling, estimated to account for about 25–35% of the rainfall in the Amazon region[10–12]. This phenomenon, together with predictions of a drier future climatic state

for the Amazon[12], may provoke a positive feedback leading to an even drier climate in the region. The combination of atmospheric drying and elevated mortality could affect ecosystem stability, which, under varying future scenarios, could lead to substantial shifts in the basic character of these forests to anything from degraded forests with lower biomass, to more radically altered systems with open canopies, or to what has been termed 'ecosystem collapse', implying a complete loss of the pre-existing forest structure and function[1,3,13,14]. Even if potential scenarios for such a tipping point have been identified, the ecological resilience of Amazon rainforests to drought, understood as the capacity of the ecosystem to function under drier conditions[15], is largely unknown.

[1]School of GeoSciences, University of Edinburgh, Edinburgh, UK. [2]Department of Geography, Faculty of Environment Society and Economy, University of Exeter, Exeter, UK. [3]School of Earth and Environmental Sciences, Cardiff University, Cardiff, UK. [4]CREAF, Cerdanyola del Vallés, Spain. [5]Research School of Biology, Australian National University, Canberra, Australian Capital Territory, Australia. [6]Instituto de Geociências, Universidade Federal do Pará, Belém, Brazil. [7]Programa de Pós-Graduação em Botânica Tropical, Museu Paraense Emílio Goeldi and Universidade Federal Rural da Amazônia, Belém, Brazil. [8]Museu Paraense Emílio Goeldi, Belém, Brazil. [9]School of Biological Sciences, University of Adelaide, Adelaide, South Australia, Australia. [10]ICREA, Barcelona, Spain. ✉e-mail: pablo.sanchez@ed.ac.uk

Evidence from drought experiments in the Amazon indicated that a primary cause of a shift in the carbon sink under prolonged drought is drought-induced tree mortality[16–19]. Indeed, even the effects of severe short-term natural drought on tree mortality have been shown to be capable of temporarily switching the sign of the regional carbon sink from positive to negative. This process could potentially trigger larger biomass loss at high drought intensity if physiological thresholds of trees are persistently surpassed[20]. One of the key physiological mechanisms contributing to mortality during drought is a decline in xylem water potential (WP) leading to embolism in the xylem vessels, restricting water flow to the tree crown and consequent transpiration[21]. Under extreme water stress, this restriction in water flow may become widespread in the xylem causing hydraulic failure, a response that is expected to increase in frequency as the Amazon becomes drier and warmer[22]. However, other processes may reduce the likelihood of these events happening at the community and ecosystem scales. Unless very severe conditions persist for a considerable time, drought-induced mortality may eventually reduce overall competition for water to the point where demand at the community scale balances the long-term reduction in rainfall. This effect would lead to eco-hydrological stabilization because any surviving tree would no longer experience drought stress and would be able to maintain its hydraulic function in the new drier climate.

Characterizing the multidecadal response to drought of an entire forest, and not just individual trees, is critical to understanding global change impacts at regional scales. However, studying whole-forest drought responses over prolonged periods is complex without long-term manipulation experiments, which are rare and costly. The Caxiuanã throughfall exclusion (TFE) experiment in the eastern Amazon[16–18,23] offers a valuable opportunity to gain insights into the impacts of sustained drought stress because it is the only tropical forest experiment where a precipitation exclusion treatment has been maintained over decades (more than 20 years) and at a large-enough scale (1 ha) to consider community-level and ecosystem-level responses (Methods).

Previous studies on this experimental site reported the mortality of individual trees related to physiological stress and probable failure of the hydraulic system, as well as cross-treatment differences in transpiration, tissue WPs and loss of xylem conductance[16,17,24–26]. In this study, we revisited the experiment to evaluate whether after more than 20 years of 50% TFE, the forest continues to experience drought stress or whether it has reached eco-hydrological stabilization under drier conditions (understood as biomass stabilization coupled with no signs of hydraulic stress). We show that despite large and occasionally precipitate declines in biomass during the first 15 years of experimental drought, the forest has maintained a stable biomass for the subsequent 7 years. While the TFE treatment initially strongly reduced the amount of water available per tree (biomass-relative water availability), the observed biomass loss during 2002–2016 subsequently led to an increase in biomass-relative water availability, equivalent to that found in the adjacent non-droughted control forest. This increase in water availability led to no statistically significant differences between the hydraulic stress measured in trees in the TFE and control forest at present. These results suggest that drought-induced biomass collapse is unlikely in this ecosystem, indicating that Amazonian rainforests can reach eco-hydrological stability after multidecadal drought, despite high mortality rates and large reductions in biomass and in the accumulation of carbon in the wood caused by prolonged severe water deficit.

## Results and discussion

### Biomass stabilized after 15 years of sustained drought

The TFE lost 85 MgC ha$^{-1}$ of aboveground biomass during the first 15 years of experimental drought (2002–2016, transition phase), which suggested a reduction of 34% of its initial biomass (248 MgC ha$^{-1}$). After the transition phase, the biomass stabilized during the following

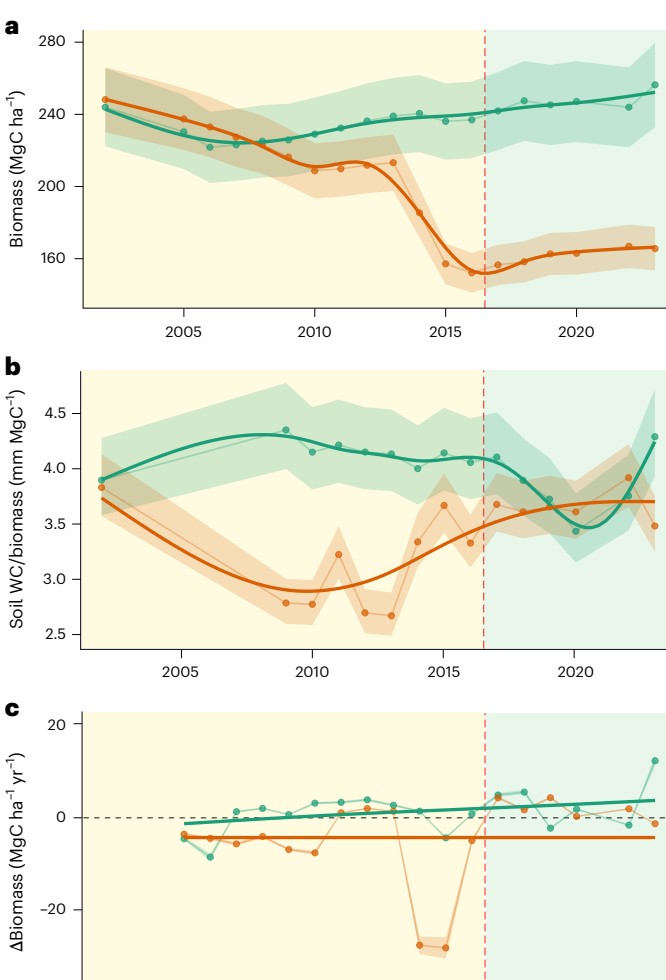

**Fig. 1 | Forest carbon time series. a–c**, Estimated aboveground wood biomass (**a**), biomass-relative water availability (**b**) and annual change in biomass (ΔBiomass) (**c**) for the control and TFE plots during the entire drought experiment period (2002–2023). General additive models were used to derive the tendency lines, which are shown as solid lines; the error bands represent the 95% confidence interval (CI) for biomass (estimated from the variability in species wood density), ΔBiomass and soil WC per unit biomass are shown. The red dashed line represents the approximate time at which the TFE plot changed from transition to stabilization phase (that is, stabilization of biomass). The two phases are also represented by the background colour. Soil water content is reported separately in Supplementary Fig. 9.

seven years (2017–2023, stabilization phase), at a mean biomass of 163.65 ± 1.47 MgC ha$^{-1}$ (mean and s.e. values given hereafter) (Fig. 1a). Biomass loss was strongly related to a disproportionately high loss of large trees during the transition phase (Supplementary Fig. 1) confirming previous studies at the same site reporting higher mortality of large trees[16,17].

The stabilization in biomass was consistent with an increase in biomass-relative water availability, calculated as the annual maximum soil water availability in the top 4 m of soil per unit biomass. During the first transition phase driven by soil moisture deficit, biomass-relative water availability dropped rapidly in the TFE forest to below 3 mm of soil water per megagram (Mg) of biomass (2.97 ± 0.07 mm MgC$^{-1}$), a reduction of 28.81% relative to that in the control plot during the same period (4.18 ± 0.03 mm MgC$^{-1}$). After 15 years of drought

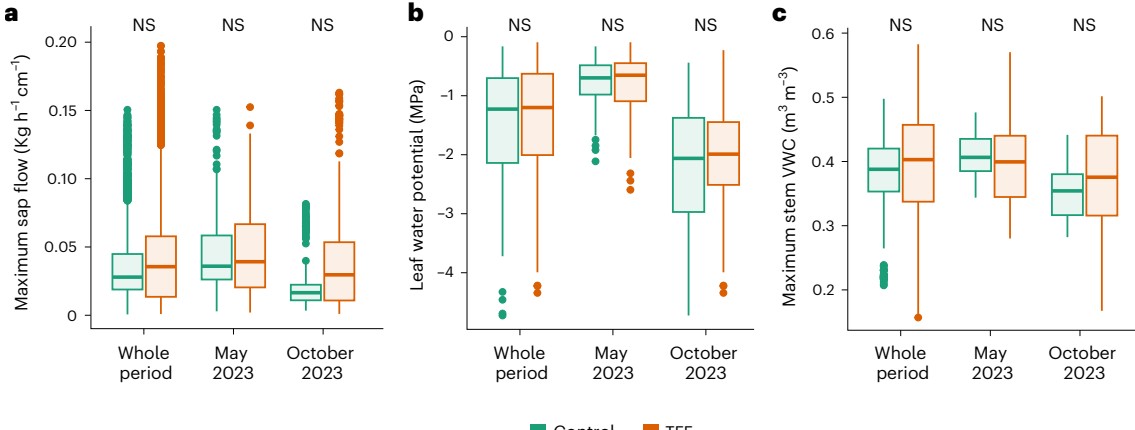

**Fig. 2 | TFE trees show hydraulic homeostasis. a–c**, Individual transpiration measured as the daily individual maximum sap flow (**a**), maximum drought stress measured as leaf WP at midday (**b**) and maximum daily stem volumetric water content (VWC) (**c**), measured in trees from the TFE and the control plots for a year (May 2023 to May 2024) and at the peak of the wet and dry season (May 2023 and October 2023, respectively). The leaf WP was measured in 352 trees, 176 from the TFE and 176 from the control plot at the peak of the wet season (May 2023) and at the peak of the dry season (October 2023). Sap flow and stem WC were monitored throughout the year in a subsample of 42 trees representative of each plot (21 trees per plot) (Supplementary Table 1 and Supplementary Figs. 3 and 5 for a month-by-month comparison). Statistical significance was tested using linear mixed models (Methods). NS, $P > 0.05$. The box plots represent the first, second and third quartiles; the whiskers represent the largest and smallest data points within $1.5 \times$ interquartile range.

treatment, the biomass stabilized, with biomass-relative water availability returning to values similar to those found in the control forest (average biomass-relative water availability of $3.89 \pm 0.07$ and $3.65 \pm 0.07$ mm MgC$^{-1}$ for the TFE and control forest, respectively). The biomass loss on the TFE clearly led to an increase in biomass-relative water availability (Fig. 1b); this was also related to an increase in growth of trees smaller than 30-cm diameter (Supplementary Fig. 2). This is consistent with their release from previous severe competition for water as predicted by recent theoretical formulations[27].

Annual change in biomass (ΔBiomass hereafter) was significantly lower in the TFE relative to the control forest during the transition phase, being negative or close to zero in most years ($-7.39 \pm 2.93$ MgC ha$^{-1}$ yr$^{-1}$ in the TFE compared to $0.17 \pm 1.12$ MgC ha$^{-1}$ yr$^{-1}$ in the control forest). Once biomass-relative water availability recovered, ΔBiomass in the TFE was mostly positive ($1.42 \pm 0.94$ MgC ha$^{-1}$ yr$^{-1}$), but still lower than observed in the control forest ($3.20 \pm 2.69$ MgC ha$^{-1}$ yr$^{-1}$) for that period (Fig. 1c). These results suggest that losses of aboveground wood biomass in the TFE were greater than gains during the transition phase; however, currently carbon gains are greater than losses, switching from a biomass carbon source to a small sink.

**Hydraulic homeostasis in trees after multidecadal drought**

Individual trees were monitored during 2023 and 2024 to test whether levels of drought stress differed between treatments. This period included very pronounced wet and dry seasons, the latter related to global warming and the very strong El Niño Southern Oscillation episode of 2023[28]. Therefore, we probably captured the two extremes of wet and dry conditions these trees are exposed to at the site. Our results showed that trees exposed to multidecadal soil drought were able to maintain their hydraulic function under the full range of conditions in both control and TFE plots. Individuals in the TFE plot showed daily transpiration rates similar to control trees throughout the year, while having a smaller drop in transpiration during the dry season (Fig. 2a and Supplementary Fig. 4). This indicated that the surviving TFE trees were able to maintain transpiration even under very dry conditions, contrasting with previous results in the same experiment during the earlier transition phase, which showed substantially reduced transpiration in the dry season in the droughted TFE plot[23,26].

TFE and control trees showed similar levels of hydraulic stress, as measured using the leaf WP at midday, both in the wet and the dry season (Fig. 2b and Supplementary Fig. 4b,d). This indicates that trees from both plots have similar access to water and the ability to transport it from roots to leaves. This is supported by the leaf WP at pre-dawn, which represents the hydraulic status of the soil and indicates similar water availability across the plots (Supplementary Fig. 4a,c). Soil water status measured during the dry season was similar to that observed in previous studies at the site during the same month[25], suggesting that the pronounced wet season in early 2023 did not have a strong impact on soil water availability at the peak of the drought. However, we cannot entirely rule out the possibility that an exceptionally wet season may have delayed the drought's effects on trees.

The similarity in current-day water status in trees from both plots was further underlined by our data, showing no differences in leaf relative water content (WC), that is, the amount of water held in leaves relative to their maximum (Supplementary Fig. 4d,f). Our results may have been affected by leaf water uptake, which may have acted to reduce the impact of water scarcity in the soil on leaf water status. However, this mechanism was previously estimated to account for about 8% of the annual transpiration at the study site[29], meaning that trees still mainly rely on soil water availability to support their hydraulic function; thus, individual trees are responsive to soil water availability.

Maximum stem water content (WC) and the reduction in stem WC from annual maxima (that is, under fully hydrated conditions during the peak of the wet season) were also similar in trees from the two plots throughout the year (Fig. 2c and Supplementary Fig. 5), as was branch volumetric WC (VWC), the WC per unit volume in branches, in the peak of the dry season of 2023 (Supplementary Fig. 6). Together, these results confirmed that throughout the year (2023–2024) trees that experienced and survived multidecadal soil drought in the TFE had similar tissue hydration to non-droughted trees in the control forest, suggesting that water was not limiting functionality differently in the TFE relative to the control forest at this point.

The trees monitored were representative of the whole plot, with the genera sampled representing 60% of the basal area in the control plot and 30% of the basal area in the TFE. These trees also covered a range of sizes, from 10 to 160-cm stem diameter (Supplementary Table 1). The reported hydraulic patterns did not change when controlling for tree size and taxonomy, either separately or together (Methods). In the case of leaf WP, for which a bigger sampling effort was possible during the peak wet and peak dry seasons of 2023, results

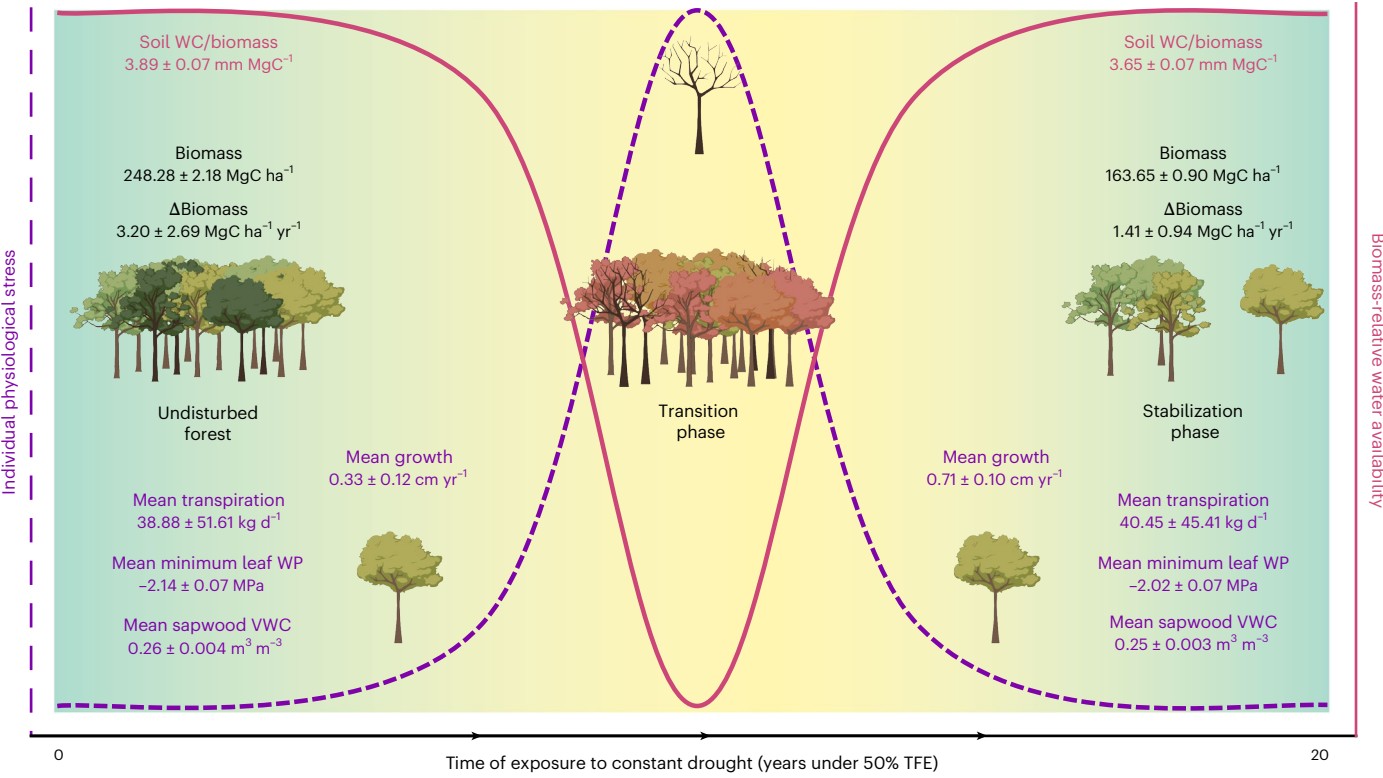

**Fig. 3 | Synthesis.** Synthetic diagram showing a summary of the results, including individual physiology (hydraulics and growth) and forest carbon dynamics. The curves showing individual physiological stress and biomass-relative water availability are approximate and simplified for plotting purposes. The data shown in the undisturbed forest refer to the control forest; the stabilization phase refers to the TFE plot from 2016 onwards. The hydraulics data shown were sampled during 2023–2024. Illustrations created with BioRender.com.

using the whole dataset (352 trees) were not different from those using only the trees that were continuously monitored (that is, the 42 trees with sap flow and stem WC data; Supplementary Fig. 4). Therefore, we are confident that our sampling design allowed us to extract general patterns and minimize confounding factors related to any effects of taxonomic identity and tree size.

Altogether, these results point towards homeostasis in the hydraulic function of surviving trees exposed to multidecadal drought. Surviving trees could transport water and maintain their hydraulic status even under the particularly dry conditions of 2023, when a more severe dry season driven by El Niño coincided with the TFE treatment. Given that adult trees have a relatively small capacity to acclimate to water deficit[25,30], this new hydraulic homeostasis was probably reached because of structural changes at the individual and ecosystem level, which together increased the biomass-relative water availability and reduced drought stress for each surviving individual tree.

### Eco-hydrological stability under drier conditions

Our data demonstrated that Amazon rainforests can persist under drier conditions, reaching eco-hydrological stability without evidence of drought stress, despite the continued implementation of a substantive experimental soil drought. Eco-hydrological stabilization arises because of a loss of biomass leading to a smaller amount of biological water demand in the ecosystem, allowing surviving trees to access enough water to secure hydraulic homeostasis and growth (Fig. 3). Given this, the resilience to drought at the forest level after prolonged exposure to drought stress probably resulted from structural changes in the ecosystem, leading to changes in resource availability that counteracted the effects of reduced soil water. The net effect was to prevent runaway biomass collapse in response to multidecadal soil drought, an outcome that might otherwise have been hypothesized based on the

high drought-related mortality rates observed earlier in this experiment and in other natural 1-year drought or experimental contexts[16,17,19,20], and based on some Earth system model predictions[31].

The plot subjected to multidecadal drought (TFE) is still forested (that is, dominated by trees with a density greater than 20%, height greater than 5 m and a leaf area index greater than 3)[32], rejecting the alternative hypothesis of ecosystem collapse and a rapid transition into a non-forested state[1,3] in response to severe soil drought conditions. Instead, the forest has transitioned to a more open canopy, with a lower number of top-canopy and emergent trees. The biomass of this forest is smaller than the mean biomass of the Amazon rainforest (266 ± 85.15 MgC ha⁻¹), but greater than the mean biomass of tropical dry forests (124.96 ± 106.65 MgC ha⁻¹) and savannas (49.00 ± 63.96 MgC ha⁻¹), located in the Amazon region (Fig. 4 and Methods)[33–36]. In fact, this forest is significantly different from tropical moist forests, tropical dry forests and tropical savannas in terms of biomass, as reported using *t*-tests (*P* < 0.001; Methods), being closer to dry forests (difference to moist forest = 103.52 MgC ha⁻¹; difference to dry forest = 38.69 MgC ha⁻¹; difference to savanna = 115.49 MgC ha⁻¹). These results were confirmed by recent biomass quantifications in the field, showing how this forest has higher values than contiguous non-forested biomes such as in the Cerrado (20.4 ± 6.5 MgC ha⁻¹) and Cerrado-Amazonian transition (32.4 ± 16.5 MgC ha⁻¹)[37], and lower values than the rainforests surrounding our study site in the North Amazonian-Guiana Shield (211.91 ± 5.03 MgC ha⁻¹)[38]. In terms of annual biomass change, after being a clear biomass carbon source during the transition, the forest reached values close to zero, which are comparable with other Amazon terra firme forests recently affected by natural drought[39].

It is important to note that our observations of biomass loss under drought could be conservative relative to the effects of large-scale

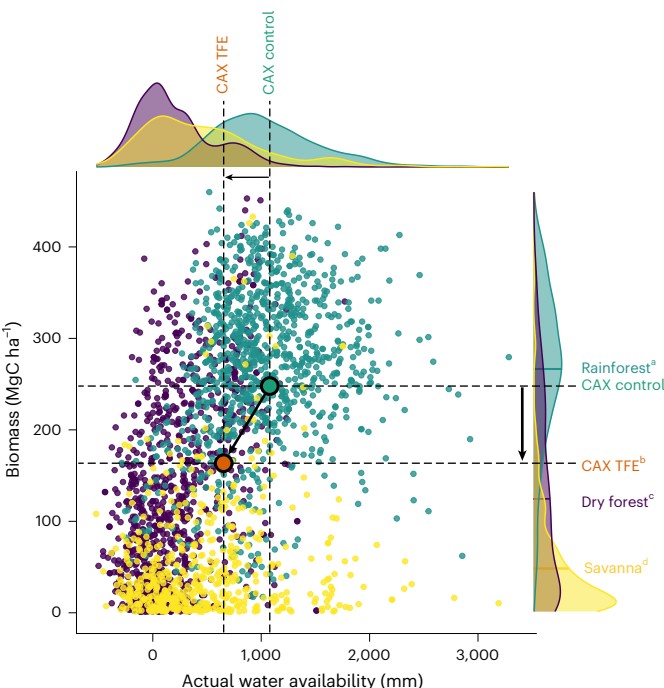

**Fig. 4 | Contextualization of biomass results in the Amazon basin.** Scatter plot and distributions of aboveground biomass and actual water availability (precipitation − evapotranspiration) for the whole Amazon basin rainforests, dry forests and savannas. The points coloured purple represent areas classified as dry forests, the green points as moist forests and the yellow points as savannas[34–36]. Control (green) and TFE (orange) data points are also shown, representing the observed slope (black arrow). Statistically significant different groups for biomass values reported using two-sided *t*-tests (*P* < 0.05) are represented by different letters (a, b, c, d). Our results showed how the TFE plot was significantly different from rainforests (*t* = 87.17, *P* < 0.001), dry forests (*t* = −62.61, *P* < 0.001) and savannas (*t* = −104.92, *P* < 0.001), while the control was not significantly different from rainforests (*t* = −1.22, *P* = 0.22) and significantly different from dry forests (*t* = −90.99, *P* < 0.001) and savannas (*t* = −126.23, *P* < 0.001).

natural drought because they only consider the effects of soil drought over one hectare, which would be amplified by increased temperature and atmospheric dryness over larger areas[40–42] as well as highly influenced by differences in edaphic characteristics, and over longer time intervals, by distance to different seed sources.

Importantly, this experiment simulated a permanent reduction in rainfall rather than the increasing variation in water availability expected in the region, and induced only soil drought, while atmospheric drought due to high vapour pressure deficit is also important. The experiment reflected the site's natural climatic variability, allowing us to examine how increased fluctuations in atmospheric moisture, particularly the extreme low values observed during the dry seasons of recent El Niño years, interact with soil drought. However, the combined effects of these factors with other stressors driven by biotic agents and disturbances such as strong winds, storms and fire[43], as well as the potential concatenation of multiple exceptional droughts, could cause Amazon rainforests to lose more biomass or require more time to stabilize than observed in our study. In this context, further work, most probably integrating ground measurements, remote sensing and modelled data, are needed to further elucidate if and how positive feedback between environmental forcing and Amazonian forests could push these ecosystems beyond their capacity for resilience.

Our work demonstrates that a substantial part of the long-term response to drought by Amazonian rainforest is not only a consequence of tree-level responses but emerges from drought-induced changes on

trees whose effects are imposed at the level of the ecosystem. While previous work showed that Amazon trees, especially larger individuals, have a low capacity to acclimate or adapt to future droughts[25,30] and that high rates of tree mortality are likely during shorter-term severe drought, we showed that forests can persist after prolonged soil drought, which may suggest it could recover and be resilient at the ecosystem level over longer timescales. This arises from negative feedback between drought-driven tree mortality and consequent changes in available soil water, which act to prevent drought-induced biomass collapse. Thus, our work also makes it clear that studies looking only at tree-level function are insufficient to understand the response of tropical forests to drought. Future analyses with a theoretical focus on ecosystem feedback coupled with long-term environmental response data, including experiments capable of capturing ecological changes at the ecosystem scale, are necessary to understand how climate extremes will affect large forested areas such as the Amazon. Importantly, the long-term eco-hydrological stability we report in this study only emerged after a highly disruptive transition phase, during which well over one-third of the biomass was lost, becoming a large biomass carbon source. If generalized over larger areas, the combined long-term results from this experiment suggest very large emissions of carbon dioxide to the atmosphere from drought-related tree mortality before eco-hydrological stabilization is re-established.

## Methods

### Site
The experimental site is located in the Caxiuanã National Forest Reserve, Pará State, Northern Brazil (1° 43′′ S, 51° 27′′ W). This site is a terra firme seasonal rainforest and receives 2,000–2,500 mm of rainfall a year, with four to six consecutive months of the year where less than 100 mm falls from July to December. The site has yellow oxisol soil[44] and a mean air temperature of approximately 25 °C.

In January 2002, a TFE experiment was constructed on 1 ha of tropical rainforest. Transparent panels were installed 1–2 m above the ground to redirect approximately 50% of the rainfall to a system of gutters. A trench was dug around the TFE to transport the water away from the experiment. A trenched control plot with no rainfall exclusion was established less than 50 m from the TFE. Consistent with some other large-scale manipulative studies, where large treatment effects are expected, the experimental treatment was not replicated for reasons of suitability, cost and maintenance, although as elsewhere pre-treatment data were recorded (see refs. 17 for more experimental details).

### Data
**Meteorological and soil moisture data.** Meteorological variables were obtained from a weather station situated in a 40-m tower located in the control forest (see Supplementary Fig. 7 for meteorological data for 2023–2024). Temperature and relative humidity were monitored using HC2S3 (Campbell Scientific) and precipitation was monitored using a tipping bucket rain gauge (TE525MM, Campbell Scientific). The vapour pressure deficit was calculated from temperature and relative humidity using the bigleaf R package[45].

Soil access pits were located in the control and TFE plots. In each soil pit, soil moisture was monitored at an hourly resolution using WC sensors (CS616, Campbell Scientific) at depths of 0, 0.5, 1, 2.5 and 4 m, which are expected to account for most of the cumulative root fraction[46]. The total WC for the first 4 m of depth was calculated for each plot, multiplying the volumetric WC for each depth segment (0–0.5, 0.5–1, 1–2.5 and 3.5–4 m) by the volume of the segment for a hectare (10,000 $m^2$ × depth of segment). The resulting absolute WC values were summed to obtain the total WC in the first 4 m of depth (l $m^{-2}$ or mm). Soil moisture time series data were available from 2008 onwards, so we assumed the first value (before TFE, that is, 2002) to be the mean for the whole period in the control forest, which was consistent with soil moisture point measurements taken during that time[47]. All meteorological

and soil moisture data were aggregated, calculating the mean per day, with the exception of precipitation, for which the sum was calculated.

**Growth and biomass data.** Since the beginning of the experiment, the stem circumference increment (growth) was measured at 1.3 m above the base for all trees with a diameter at breast height (DBH) > 10 cm using dendrometer bands, according to previous implementations[48,49]. Measurements were taken quarterly since the beginning of the experiment, at the start of January, April, July and October, with the exception of 2008 and 2021, when data collection was not possible. For trees with buttresses, DBH was measured above the buttress using a ladder at a permanently marked location. Growth measurements higher or lower than three s.d. from the mean for each individual tree were removed. Individual growth is presented as stem increment (cm) per year. DBH was calculated from circumference growth by summing the current year growth divided by pi to the previous year DBH.

To estimate individual aboveground biomass from the DBH, we used allometric models, based on previous implementations in the same site[16]. First, the following allometric equation was used to calculate tree height from the DBH[50]:

$$\text{Height (m)} = 227.35 \left(1 - \exp\left(-0.139\,\text{DBH}^{0.5550}\right)\right)$$

Then, we applied a recently derived allometric equation specifically fitted for the site[51] using the measured DBH (cm), the previously estimated height (H, m) and wood density (WD, g cm$^{-3}$) taken from the literature[52]. This allometric equation took the following form:

$$\text{Aboveground biomass (kg)} = 0.088 \left(\text{WD H (DBH)}^2\right)^{0.954}$$

To account for uncertainty in wood density estimates, we calculated aboveground biomass for the higher and lower 95% CIs using the wood density s.e. Individual aboveground biomass was then aggregated per plot and according to year, summing individual tree biomass (both for mean wood density and the higher and lower 95% CIs). Mean aboveground biomass per plot was also calculated according to the same procedure but using the mean instead of the sum. ΔBiomass was calculated as the difference in aboveground biomass between consecutive years, divided by the number of years of difference between them. ΔBiomass was reported for biomass using mean wood density and the higher and lower 95% CIs. We also calculated the WC per unit biomass by dividing the maximum annual absolute WC in the first 4 m of depth by the annual biomass (using mean wood density and higher and lower 95% CIs), which we interpreted as a measure of biomass-relative water availability.

Biomass patterns were very similar to the one reported in ref. 16 for 2002–2015, during which a 40% biomass loss in the TFE plot was reported. This variation in biomass estimates can be related to the fact that, in the previous work, diameters were measured using measuring tape while in the current study we used dendrometer bands, which were used more continuously, especially after 2015 (Methods). We also report biomass calculated using DBH measuring tapes (Supplementary Fig. 8); the estimated loss in biomass during the transition phase was of 84 MgC ha$^{-1}$ (only 1 MgC ha$^{-1}$ difference with the results reported in the main text) and in which the stabilization phase was also apparent from 2016 onwards.

**Sap flow data.** Sap flow for a total of 42 individual trees (21 in the control forest and 21 in the TFE) was monitored using EMS81 systems (environmental monitoring system (EMS); http://www.emsbrno.cz), which retrieved the whole-tree sap flow by using the heat balance method[53] at a resolution of 15 min from May 2023 to May 2024. This sensor design is suitable for trees, has proved robust to the tropical environment and has previously been used with success to quantify sap flux at this site[24,26]. Eight small trees (<30-cm DBH), eight medium trees (>30-cm DBH and <60-cm DBH) and five big trees (>60-cm DBH) were sampled in each plot; when possible, species from the same genus were replicated in both plots (Supplementary Table 1). Sap flow per unit stem circumference (kg h$^{-1}$ cm$^{-1}$) was also calculated by applying the following equation provided by the EMS:

$$\text{SF}_{\text{area}} = \frac{\text{SF}_{\text{tree}}}{A - (2\text{pi}\,B)}$$

Where A is the tree circumference (cm) and B is the bark thickness (cm), which was measured following the standard procedure described by the EMS.

Sensors were installed in May 2023. To obtain the baseline sap flow data, we performed a 10% quantile regression for each individual tress using the rq function of the package quantreg[54], setting hourly sap flow as a response variable and time as a predictor variable, and setting the tau argument to 0.1. We then used each individual model to predict values for the time period. Finally, we subtracted the predicted values from the raw values.

We aggregated the hourly sap flow data to daily data by calculating the 90% quantile and the sum. We also calculated the percentage of reduction in maximum daily sap flow relative to the annual maximum per individual tress as a measure of reduction in maximum daily transpiration over total capacity, which quantifies the regulation of transpiration. In the main text, we refer to these results as transpiration, according to previous studies using sap flow sensors[24,26].

**Leaf WP and relative WC data.** To characterize individual hydraulic status and hydraulic stress, we measured leaf water potential (WP) for 352 trees (176 trees per plot, representing 35% and 38% of the control forest and TFE plots, accounting for 59% and 34% of the basal area, respectively). Measurements were taken at pre-dawn (from 4:00 to 6:00), when leaves are in equilibrium with soil water and at midday (from 11:45 to 14:00), when hydraulic stress is expected to be close to its maximum. One sun-exposed branch from the top of the crown was sampled in each case, and measurements were taken in a minimum of two leaves per branch. Measurements were taken in the field immediately after collection using a pressure chamber (model 1505D, PMS Instruments; 0.05-MPa resolution). Leaf WP was measured at the peak of the wet season (May 2023) and at the peak of the dry season (October 2023).

We performed extra leaf measurements for the 42 trees for which we were monitoring sap flow (Supplementary Table 1). For these trees, we measured the WP of leaves obtained from three different branches from the top of the crown. WPs were taken in two leaves per branch. For each one of the three branches sampled, three to five leaves were bagged, minimizing the amount of air in the bag; they were stored in a dark isolated compartment, transported to the field station laboratories and weighed ($M_{\text{fresh}}$) using a precision balance (±0.1 mg). Right after each measurement, leaves were submerged in water for 12 h and reweighed for turgid mass at full hydrated state ($M_{\text{turgid}}$). Finally, leaves were oven-dried for a minimum of 24 h at 70 °C and reweighed for dry mass ($M_{\text{dry}}$). Relative water content (RWC) was calculated using the following equation:

$$\text{RWC} = \frac{M_{\text{fresh}} - M_{\text{dry}}}{M_{\text{turgid}} - M_{\text{dry}}}$$

Leaf WP and RWC at pre-dawn and midday for these 42 trees were measured at the peak of the wet season (May 2023), the beginning of the dry season (July 2023) and at the peak of the dry season (October 2023).

**Branch wood volumetric WC data.** During the peak of the dry season (October 2023), we complemented leaf measurements with branch

volumetric water content (VWC) at pre-dawn and midday for the 42 monitored trees. Two twig segments were cut from each branch collected for leaf WP, measuring approximately 5–12 mm in diameter (mean = 6.3 mm) and 30–50 mm long (mean = 42.1 mm). One segment was used to determine sapwood WC as follows: the branch diameter was measured with and without bark using vernier gauge callipers (±0.02 mm); the segment without bark was weighed (fresh mass, $M_{fresh}$) using a precision balance (±0.1 mg); volume ($V_{fresh}$) was measured using mass balance volume displacement; when present, the heartwood or pith diameter was measured; the segment was oven-dried for a minimum of 48 h and reweighed for dry mass ($M_{dry}$). The VWC was determined as:

$$VWC = \frac{M_{fresh} - M_{dry}}{V_{fresh}}$$

**Stem volumetric WC data.** We installed frequency domain reflectometry sensors to measure WC in the 42 monitored trees (Supplementary Table 1) at a resolution of 15 min according to the method described in ref. 55. To do this, we first selected a position sheltered from the sun, when possible, and then carefully removed the bark from a small area at breast height. We then drilled parallel holes into the sapwood making sure that the drill bits were the same length as the sensor needles. To maximize accuracy during this process, we used a drill guide. The 3-mm drill bit was slightly thinner than the sensor needles (3.175 mm), assuring close contact between the sensor and the woody tissue. Once the sensor was in place, it was sealed using a silicon-based sealant to ensure no interaction with the atmosphere. Finally, we covered all sensors with solar radiation shields (for a detailed description of the whole process, see https://github.com/lionmartius/Splish-Splash-Sap). Sensors were installed in May 2023.

Data were calibrated using tropical tree calibration[55]; temperature correction was applied by calculating the difference in temperature from the mean, then multiplying it by the temperature effect reported in a previous study at the same site[55] and subtracting this value from the measured values:

$$\text{Temperature corrected VWC} = VWC - T_{diff} \times b$$

where $T_{diff}$ is the difference between measured and annual mean temperature (reference point) and $b$ is the temperature effect (−0.000974).

**Amazon basin data.** We obtained global products on biomass[33], precipitation[56] and evapotranspiration[57] data and cropped them to keep only the Amazon basin area. We also obtained a land cover map for the Amazon basin[34,35] and a delimitation of the dry forest areas in South America (data obtained from http://www.dryflor.info/data)[36] (all data were accessed in July 2024). After homogenizing raster projections using the raster R package[58], we sampled 10,000 coordinates in areas classified as rainforest, 10,000 coordinates in areas classified as dry forests and 10,000 coordinates in areas classified as savanna throughout the Amazon basin. We ensured that the distribution of the sampled coordinates was similar to the one shown for the whole rainforest, dry forest and savanna areas (that is, the same range and mean). We then extracted biomass, precipitation and evapotranspiration data for the sampled coordinates. Finally, we calculated actual water availability by subtracting evapotranspiration from precipitation, as a proxy of water available in the soil, following previous implementations of water balance, assuming a negligible effect of surface run-off and deep drainage[59].

**Statistical analyses**
All statistical analyses were conducted in R[60]. All response variables were log-transformed to improve the normal distribution of linear model residuals, except for the variables representing reduction in transpiration and stem WC, which were transformed using the square root transformation. With regard to negative variables (leaf WPs), the absolute value was calculated before transformation. Linear mixed models were performed using the lme4 R package[61], and linear models using the stats package. All spatially explicit data were processed using the raster, terra and sf R packages[58,62,63].

**Statistical analyses of plant hydraulics.** To test whether drought still had an impact on the hydraulic function of individual trees, we compared hydraulic measurements taken in trees exposed to multidecadal drought and trees under normal conditions during 2023–2024.

To assess the differences between plots in individual maximum daily sap flow and daily sap flow reduction from the annual maxima, we used linear mixed models, including plot and diameter as a fixed effect, and individual nested within genus as a random effect. We also implemented models excluding genus and diameter; the results converged. Comparisons were made for the whole period (May 2023 to May 2024) and separately for each month (Fig. 2a and Supplementary Fig. 4).

To assess differences between plots in individual leaf WP, we used linear mixed models, including these as response variables, plot and diameter as the fixed effect and genus as the random effect. Comparisons were made for the high-density sampling campaigns (n = 352, May and October 2023) and for campaigns performed in monitored trees (n = 42, May, July and October 2023) (Fig. 2a and Supplementary Fig. 4). When analysing monitored trees, the random effect was modified to include individual nested within genus because three branches per individual were measured. Differences in leaf WC and branch WC between plots were also analysed using mixed models, including plot and diameter as the fixed effect and individual nested within genus as the random effect (Supplementary Fig. 4). In all cases, we also implemented models excluding genus or diameter; the results converged.

To assess the differences between plots in individual stem WC and WC reduction, we used linear mixed models, including plot and diameter as the fixed effect and individual nested within genus as the random effect. We also implemented models excluding genus or diameter; the results converged. Comparisons were made for the whole period (May 2023 to May 2024) and separately for each month (Fig. 2c and Supplementary Fig. 5).

The effects of size and taxonomy on the hydraulic variables were evaluated independently by quantifying the variance explained by genus and DBH using linear models (Supplementary Table 2). Diameter had a very small effect on hydraulic variables, whereas genus had a larger effect. However, results including and excluding diameter and genus separately and together converged, demonstrating that even if genus may have a strong predictive power, it does not affect the direction or the magnitude of the plot effect, which is the one we are interested in. However, future studies with higher replication at the genus level will help further evaluation of taxonomic and phylogenetic effects on Amazonian tree hydraulics; this is beyond the scope of the current work, in which we focused on differences between plots and not on the specific drivers of variability in hydraulic metrics.

**Statistical analyses of tree density and biomass.** To better understand the drivers of forest biomass changes, we tested for relationships between tree density and biomass. We were interested in the effects of total tree density, emergent and top-canopy tree density, and subcanopy and low canopy tree density on plot total and mean biomass. To do so, we first classified trees into two groups related to their size and exposure to the top of the canopy: top-canopy and emergent trees, with a DBH greater than 30 cm and subcanopy trees, with a DBH smaller than 30 cm (equivalent to a height of around 30 m, as reported by the allometric function described above, this being the mean tree canopy height of the control plot of 22 m for all trees with a DBH greater than 10 cm). Then, we used linear models, including annual forest biomass as a response variable and tree density (either

total, emergent or subcanopy), as an explanatory variable, using data from the TFE and control forest separately. To account for the effects of temporal autocorrelation, we also included year as a covariate, which did not change the significance of the results (Supplementary Fig. 1).

**Statistical analyses of tree growth.** To complement the hydraulic results on evaluating differences in tree physiology, we also analysed differences in individual growth between plots. To do so, we used mixed models, including annual individual growth as a response variable and plot as a fixed effect, while setting individual nested within genus as the random effect. We report the plot effect on individual growth for emergent and subcanopy trees (see the previous section), showing how subcanopy trees tend to present higher growth after the stabilization phase (Supplementary Fig. 2).

Then, we tested whether the changes in growth were related to biomass-relative water availability. To test this, we used linear models, including mean growth for emergent and subcanopy trees as a response variable and WC per unit of biomass as an explanatory variable, using data from the TFE and control forest, separately. To account for the effects of temporal autocorrelation, we also included year as a covariate, which did not change the significance of the results (Supplementary Fig. 1). We reported how higher biomass-relative water availability was related to higher mean growth in the TFE plot.

**Statistical analyses of whole Amazon basin datasets.** To better contextualize our results within the whole Amazon basin, we positioned the two plots (control forest and TFE) in a scatter plot, showing the relationship between biomass and actual water availability (precipitation − evapotranspiration) for the forests of the Amazon basin, including rainforest, dry forest and savanna. To facilitate visualization, only 1,000 points per group (rainforest, dry forest and savanna) were plotted. In the Caxiuanã plots (control forest and TFE), we used the annual soil WC (the mean for 2017–2023) as a measure of hydrological water availability. Then, we used $t$-tests in the stats R package to evaluate whether the two plots were significantly different from the biomass distributions, previously log-transformed, for rainforest, dry forest and savanna. Our results showed how the TFE plot was significantly different from rainforests ($t = 87.17$, $P < 0.001$), dry forests ($t = −62.61$, $P < 0.001$) and savannas ($t = −104.92$, $P < 0.001$), whereas the control forest was not significantly different from rainforests ($t = −1.22$, $P = 0.22$), this being significantly different from dry forests ($t = −90.99$, $P < 0.001$) and savannas ($t = −126.23$, $P < 0.001$).

### Reporting summary

Further information on research design is available in the Nature Portfolio Reporting Summary linked to this article.

## Data availability

The minimum dataset needed to replicate the analyses is available via figshare at https://doi.org/10.6084/m9.figshare.27960801 (ref. 64).

## Code availability

The code used is available via GitHub at https://github.com/pablo-sanchezmart/Sanchez-Martinez-etal-2025-Amazon-rainforest-ecohydrological-adjustment (ref. 65).

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

## Acknowledgements

We thank the DRYFLOR network (http://www.dryflor.info/) for making their data freely available. This research was supported by a Natural Environment Research Council (NERC) grant no. NE/W006308/1 to P.M. and L.R., a Royal Society Wolfson Fellowship no. RSWF\211008 to P.M., the Newton Fund through a Met Office Climate Science for Service Partnership Brazil (CSSP Brazil) grant to L.R. and P.M., and an NERC independent fellowship grant no. NE/N014022/1 to L.R.

## Author contributions

P.S.-M., L.R.M., P.B. and P.M. designed the study. P.S.-M., L.R.M., P.B., M.S., O.B., I.C., V.N.-R., J.A.S., A.C.D.C. L.R., M.M. and P.M. provided the

data and helped develop the framework. P.S.-M. analysed the data and wrote the first draft of the manuscript. L.R.M., P.B., M.S., O.B., V.N.-R., L.R., M.M., R.S., S.R. and P.M. contributed substantially to manuscript revisions.

## Competing interests

The authors declare no competing interests.

## Additional information

**Correspondence and requests for materials** should be addressed to Pablo Sanchez-Martinez.

# Reporting Summary

## Statistics

For all statistical analyses, confirm that the following items are present in the figure legend, table legend, main text, or Methods section.

| n/a | Confirmed | |
|---|---|---|
| ☐ | ☒ | The exact sample size ($n$) for each experimental group/condition, given as a discrete number and unit of measurement |
| ☐ | ☒ | A statement on whether measurements were taken from distinct samples or whether the same sample was measured repeatedly |
| ☐ | ☒ | The statistical test(s) used AND whether they are one- or two-sided<br>*Only common tests should be described solely by name; describe more complex techniques in the Methods section.* |
| ☐ | ☒ | A description of all covariates tested |
| ☐ | ☒ | A description of any assumptions or corrections, such as tests of normality and adjustment for multiple comparisons |
| ☐ | ☒ | A full description of the statistical parameters including central tendency (e.g. means) or other basic estimates (e.g. regression coefficient) AND variation (e.g. standard deviation) or associated estimates of uncertainty (e.g. confidence intervals) |
| ☐ | ☒ | For null hypothesis testing, the test statistic (e.g. $F$, $t$, $r$) with confidence intervals, effect sizes, degrees of freedom and $P$ value noted<br>*Give P values as exact values whenever suitable.* |
| ☒ | ☐ | For Bayesian analysis, information on the choice of priors and Markov chain Monte Carlo settings |
| ☒ | ☐ | For hierarchical and complex designs, identification of the appropriate level for tests and full reporting of outcomes |
| ☒ | ☐ | Estimates of effect sizes (e.g. Cohen's $d$, Pearson's $r$), indicating how they were calculated |

*Our web collection on statistics for biologists contains articles on many of the points above.*

## Software and code

Policy information about availability of computer code

| Data collection | R programming software, Microsoft Excel |
|---|---|
| Data analysis | R programming software |

For manuscripts utilizing custom algorithms or software that are central to the research but not yet described in published literature, software must be made available to editors and reviewers. We strongly encourage code deposition in a community repository (e.g. GitHub). See the Nature Portfolio guidelines for submitting code & software for further information.

## Data

Policy information about availability of data

All manuscripts must include a data availability statement. This statement should provide the following information, where applicable:
- Accession codes, unique identifiers, or web links for publicly available datasets
- A description of any restrictions on data availability
- For clinical datasets or third party data, please ensure that the statement adheres to our policy

The minimum dataset needed to replicate the analyses has been published in fighsare.

# Research involving human participants, their data, or biological material

Policy information about studies with human participants or human data. See also policy information about sex, gender (identity/presentation), and sexual orientation and race, ethnicity and racism.

| | |
|---|---|
| Reporting on sex and gender | Does not apply to this study. |
| Reporting on race, ethnicity, or other socially relevant groupings | Does not apply to this study. |
| Population characteristics | Does not apply to this study. |
| Recruitment | Does not apply to this study. |
| Ethics oversight | The University of Edinburgh |

Note that full information on the approval of the study protocol must also be provided in the manuscript.

# Field-specific reporting

Please select the one below that is the best fit for your research. If you are not sure, read the appropriate sections before making your selection.

☐ Life sciences      ☐ Behavioural & social sciences      ☒ Ecological, evolutionary & environmental sciences

For a reference copy of the document with all sections, see nature.com/documents/nr-reporting-summary-flat.pdf

# Ecological, evolutionary & environmental sciences study design

All studies must disclose on these points even when the disclosure is negative.

| | |
|---|---|
| Study description | In January 2002, a through-fall exclusion experiment (TFE) was constructed on one hectare of tropical rain forest. Transparent panels were installed 1-2 meters above the ground to redirect approximately 50% of the rainfall to a system of gutters. A trench was dug around the TFE to transport the water away from the experiment. A trenched control plot with no rainfall exclusion was established < 50 m from the TFE. |
| Research sample | About 1000 tropical trees were monitored in the field during the study. |
| Sampling strategy | Since the beginning of the experiment, stem circumference increment (growth) was measured at 1.3 meters above the base for all trees with DBH >10cm using dendrometer bands, following previous implementations. Measurements were taken quarterly since the beginning of the experiment, at the start of January, April, July and October, with the exception of 2008 and 2021, when data collection was not possible. For trees with buttresses, DBH was measured above the buttress using a ladder at a permanently-marked location. Growth measurements greater or lower than three standard deviations from the mean for each individual were removed. Individual growth is presented as stem increment (cm) per year. DBH was calculated from growth by summing the current year growth divided by pi to the previous year DBH. |
| Data collection | Measurements were taken every three months by the project research assistants. |
| Timing and spatial scale | The site was monitored from 2002 until the present, involivng two one-hectare plots in the Amazon rain forest. |
| Data exclusions | No data was excluded. |
| Reproducibility | Methods applied in this study are widely used and reproduced in the literature (see methods section). |
| Randomization | All adult individuals (DBH > 10) were sampled for biomass and productivity. |
| Blinding | Data was measured by field assistants not directly involved in the production of the scientific articles. |

Did the study involve field work?      ☒ Yes      ☐ No

## Field work, collection and transport

| | |
|---|---|
| Field conditions | This site is a terra firme seasonal rain forest and receives 2000-2500 mm of rainfall a year, with six consecutive months of the year where less than 100 mm falls from July to December. The site has yellow oxisol soil 44 and a mean air temperature of c. 25°C. |

| Location | The experimental site is located in the Caxiuanã National Forest Reserve, Pará State in Northern Brazil (1°43'S, 51°27'W). |
|---|---|
| Access & import/export | When working in the plots, people always try to avoid walking out of the main paths to minimize impact on the forest. Researchers and technicians always brought all the garbage generated in the field back to the research station and all the work was designed so it minimized the impact on the surrounding areas. |
| Disturbance | The drought experiment had an impact on the forest. This impact was minimized by only applying experimental drought to 1 hectare of forest (i.e., non replicated). The droughted forest was monitored very intensely to make sure the highest amount of information was taken, minimizing the need to repeat this experiment in similar areas. |

# Reporting for specific materials, systems and methods

We require information from authors about some types of materials, experimental systems and methods used in many studies. Here, indicate whether each material, system or method listed is relevant to your study. If you are not sure if a list item applies to your research, read the appropriate section before selecting a response.

## Materials & experimental systems

| n/a | Involved in the study |
|---|---|
| ☐ | ☐ Antibodies |
| ☐ | ☐ Eukaryotic cell lines |
| ☐ | ☐ Palaeontology and archaeology |
| ☐ | ☐ Animals and other organisms |
| ☐ | ☐ Clinical data |
| ☐ | ☐ Dual use research of concern |
| ☐ | ☒ Plants |

## Methods

| n/a | Involved in the study |
|---|---|
| ☒ | ☐ ChIP-seq |
| ☒ | ☐ Flow cytometry |
| ☒ | ☐ MRI-based neuroimaging |

## Antibodies

| Antibodies used | *Describe all antibodies used in the study; as applicable, provide supplier name, catalog number, clone name, and lot number.* |
|---|---|
| Validation | *Describe the validation of each primary antibody for the species and application, noting any validation statements on the manufacturer's website, relevant citations, antibody profiles in online databases, or data provided in the manuscript.* |

## Eukaryotic cell lines

Policy information about cell lines and Sex and Gender in Research

| Cell line source(s) | *State the source of each cell line used and the sex of all primary cell lines and cells derived from human participants or vertebrate models.* |
|---|---|
| Authentication | *Describe the authentication procedures for each cell line used OR declare that none of the cell lines used were authenticated.* |
| Mycoplasma contamination | *Confirm that all cell lines tested negative for mycoplasma contamination OR describe the results of the testing for mycoplasma contamination OR declare that the cell lines were not tested for mycoplasma contamination.* |
| Commonly misidentified lines (See ICLAC register) | *Name any commonly misidentified cell lines used in the study and provide a rationale for their use.* |

## Palaeontology and Archaeology

| Specimen provenance | *Provide provenance information for specimens and describe permits that were obtained for the work (including the name of the issuing authority, the date of issue, and any identifying information). Permits should encompass collection and, where applicable, export.* |
|---|---|
| Specimen deposition | *Indicate where the specimens have been deposited to permit free access by other researchers.* |
| Dating methods | *If new dates are provided, describe how they were obtained (e.g. collection, storage, sample pretreatment and measurement), where they were obtained (i.e. lab name), the calibration program and the protocol for quality assurance OR state that no new dates are provided.* |

☐ Tick this box to confirm that the raw and calibrated dates are available in the paper or in Supplementary Information.

| Ethics oversight | *Identify the organization(s) that approved or provided guidance on the study protocol, OR state that no ethical approval or guidance was required and explain why not.* |
|---|---|

Note that full information on the approval of the study protocol must also be provided in the manuscript.

# Animals and other research organisms

Policy information about studies involving animals; ARRIVE guidelines recommended for reporting animal research, and Sex and Gender in Research

| | |
|---|---|
| Laboratory animals | *For laboratory animals, report species, strain and age OR state that the study did not involve laboratory animals.* |
| Wild animals | *Provide details on animals observed in or captured in the field; report species and age where possible. Describe how animals were caught and transported and what happened to captive animals after the study (if killed, explain why and describe method; if released, say where and when) OR state that the study did not involve wild animals.* |
| Reporting on sex | *Indicate if findings apply to only one sex; describe whether sex was considered in study design, methods used for assigning sex. Provide data disaggregated for sex where this information has been collected in the source data as appropriate; provide overall numbers in this Reporting Summary. Please state if this information has not been collected. Report sex-based analyses where performed, justify reasons for lack of sex-based analysis.* |
| Field-collected samples | *For laboratory work with field-collected samples, describe all relevant parameters such as housing, maintenance, temperature, photoperiod and end-of-experiment protocol OR state that the study did not involve samples collected from the field.* |
| Ethics oversight | *Identify the organization(s) that approved or provided guidance on the study protocol, OR state that no ethical approval or guidance was required and explain why not.* |

Note that full information on the approval of the study protocol must also be provided in the manuscript.

# Clinical data

Policy information about clinical studies
All manuscripts should comply with the ICMJE guidelines for publication of clinical research and a completed CONSORT checklist must be included with all submissions.

| | |
|---|---|
| Clinical trial registration | *Provide the trial registration number from ClinicalTrials.gov or an equivalent agency.* |
| Study protocol | *Note where the full trial protocol can be accessed OR if not available, explain why.* |
| Data collection | *Describe the settings and locales of data collection, noting the time periods of recruitment and data collection.* |
| Outcomes | *Describe how you pre-defined primary and secondary outcome measures and how you assessed these measures.* |

# Dual use research of concern

Policy information about dual use research of concern

## Hazards

Could the accidental, deliberate or reckless misuse of agents or technologies generated in the work, or the application of information presented in the manuscript, pose a threat to:

| No | Yes | |
|---|---|---|
| ☒ | ☐ | Public health |
| ☒ | ☐ | National security |
| ☒ | ☐ | Crops and/or livestock |
| ☒ | ☐ | Ecosystems |
| ☒ | ☐ | Any other significant area |

## Experiments of concern

Does the work involve any of these experiments of concern:

| No | Yes | |
|----|-----|---|
| ☒ | ☐ | Demonstrate how to render a vaccine ineffective |
| ☒ | ☐ | Confer resistance to therapeutically useful antibiotics or antiviral agents |
| ☒ | ☐ | Enhance the virulence of a pathogen or render a nonpathogen virulent |
| ☒ | ☐ | Increase transmissibility of a pathogen |
| ☒ | ☐ | Alter the host range of a pathogen |
| ☒ | ☐ | Enable evasion of diagnostic/detection modalities |
| ☒ | ☐ | Enable the weaponization of a biological agent or toxin |
| ☒ | ☐ | Any other potentially harmful combination of experiments and agents |

# Plants

**Seed stocks**

Des not apply.

**Novel plant genotypes**

Des not apply.

**Authentication**

Des not apply.

