## [Peer Review File · Nature Ecology & Evolution]

Amazon rainforest adjusts to long-term experimental drought

Corresponding Author: Dr Pablo Sanchez-Martinez

Version 0:

Decision Letter:

28th January 2025

Dear Dr Sanchez-Martinez,

Your manuscript entitled "The fate of Amazon rain forests under drought: collapse or stabilisation?" has now been seen by two reviewers. As you will see from their reports, the reviewers find your work of considerable potential interest and have made a number of comments to improve the paper. Therefore, we have decided to invite a revision for further consideration.

We are committed to providing a fair and constructive peer-review process. Do not hesitate to contact us if there are specific requests from the reviewers that you believe are technically impossible or unlikely to yield a meaningful outcome. Please note that we concur with the reviewers' view that the conclusions regarding drought-induced ecosystem collapse (or lack thereof) should be toned down, starting from the title.

* Please highlight all changes in the manuscript text file and provide it Microsoft Word format with line numbers.

* If you have not done so already please begin to revise your manuscript so that it conforms to our Article format instructions at <http://www.nature.com/natecolevol/info/final-submission>. Refer also to any guidelines provided in this letter.

* Extended Data Figures: please ensure that any supplementary figures and tables that are crucial to the manuscript's conclusions are converted into Extended Data figures and tables to increase visibility of these data. Extended Data figures and tables are online-only (present in the online PDF and full-text HTML versions of the paper), peer-reviewed display items that provide essential background to the article but are not included in the main article due to space constraints. A maximum of ten Extended Data display items (figures and tables) is permitted.

Once ready, please use the link below to submit your revised manuscript and related files:

Link Redacted

We hope to receive your revised manuscript within two months weeks. If you cannot send it within this time, please let us know. We will be happy to consider your revision so long as nothing similar has been accepted for publication at Nature Ecology & Evolution or published elsewhere.

Nature Ecology & Evolution is committed to improving transparency in authorship. As part of our efforts in this direction, we are now requesting that all authors identified as 'corresponding author' on published papers create and link their Open Researcher and Contributor Identifier (ORCID) with their account on the Manuscript Tracking System (MTS), prior to acceptance. ORCID helps the scientific community achieve unambiguous attribution of all scholarly contributions. You can create and link your ORCID from the home page of the MTS by clicking on 'Modify my Springer Nature account'. For more information please visit

please visit www.springernature.com/orcid.

Please do not hesitate to contact me if you have any questions or would like to discuss these revisions further. Thank you for the opportunity to consider your work.

[redacted]

Reviewers' comments:

Reviewer #1 (Remarks to the Author):

This study addresses a very consequential theme -- the fate of the Amazon under increased drought. It uses a 20-year throughfall reduction experiment to show that although the drought treatment initially reduced biomass dramatically, the droughted forest reached a new lower-biomass equilibrium with various metrics of hydraulic stress similar to those observed in the control plot. This demonstrates the capacity of forests in this part of the Amazon to equilibrate to dramatic reductions in water availability, suggesting that collapse into a non-forested state is unlikely. This is a valuable insight that advances our understanding of Amazonian forest responses to drought.

Overall, the research methodology is solid, and the paper is well-written, and my concerns are mostly minor.

I do have one general concern: in places, the interpretation goes beyond what I believe to be warranted, in large part because there is limited acknowledgment of the differences between the throughfall exclusion experiment and expected future drought conditions. Importantly, the experiment (1) induced a one-time permanent change, as opposed to the large (and increasing) variability in water availability expected in the future, and (2) induced only soil drought, whereas atmospheric drought (high VPD) is an extremely important component of natural drought and is coupled to soil drought. While there is some good discussion of the latter in the paragraph starting on line 237, the former is not discussed, and both seem a bit under-recognized throughout the manuscript. Specifically, I felt that the title and text on line 109 ("These results rule out drought-induced biomass collapse...") were particularly misleading. The language of the entire manuscript should be carefully reviewed to ensure that the interpretation is consistent with the reality that the experiment does not perfectly mimic natural drought.

Specific comments:

Title – Based on the title, I expected the study to give predictions for the Amazon under climate change (increasing drought). While it certainly helps inform our understanding of the Amazon's future under drought, it falls short of re-creating the future drought conditions in the Amazon.

Line 45 – define "net wood productivity" is not a conventional term, and most closely approximates (aboveground) woody productivity, a component of net primary productivity that would never be negative. I'd suggest using something like Δ biomass or Δ AGB. (Net ecosystem productivity is also inappropriate because it includes components other than biomass.)

Line 49- can remove "is considered to be"

Line 67 – reword "poorly unknown"

Lines 109-113- the language "rule out" goes beyond what can reasonably be inferred from this experiment, as described above.

Line 124 – biomass relative water availability – (this may be somewhat biased because big trees would have a larger proportion of heartwood)

Line 138 – define "net aboveground productivity". This is not a conventional term.

Line 258 – fix grammar

Line 295- change 1st sentence to past tense

Fig. 1 – Use a different term for "Productivity", such as Δ Biomass. "Productivity" would most commonly refer to gross or net primary productivity (or one of their components) and would not be negative (except for net ecosystem productivity).

Reviewer #2 (Remarks to the Author):

This manuscript is well-written, interesting, and timely. The study suggests that after two decades of drought and substantial drought-induced tree mortality, a one-hectare plot of Amazonian rainforest reached ecohydrological stability. The authors reached this conclusion because the largest trees had died meaning that more water was available for the remaining trees. The authors found that the remaining living trees in the experimental plot had leaf water potential, sap flow, and tissue water content similar to the trees in the control plot. These are very interesting findings with broad implications, especially because recent studies show that the Amazonian forest could reach a tipping point and an ecosystem collapse.

The features of this manuscript that I consider to be of immediate interest are:

- The results that smaller trees (<30 cm diameter) increased their growth during what the authors are calling the stabilization phase is interesting, however, these trees will likely not be able to become very big as the larger trees in the drought experiment died. As the authors describe, these findings are interesting because they shed light on what this forest may look like under future drier climates.
- It is also of immediate interest that once water availability increased after reaching ecohydrological stability productivity also increased, however, it continued to be lower than the control plot. Meaning that the carbon uptake of this forest under a drier climate will be lower than the current levels of carbon uptake.
- During the period of ecohydrological stability there was a strong El Niño episode in 2023 and the forest in the drought

experiment remained stable, suggesting that drought-induced biomass collapse may not have occurred in this ecosystem under drier conditions. However, this does not necessarily rule this out and please see my comment below.

In lines 109-113, the authors state that their findings rule out drought-induced biomass collapse in this ecosystem; however, this may not be the case if there are more severe droughts like the one in 2014-2015. These results are promising but to say that they "rule out" that this ecosystem will not suffer from biomass collapse due to drought seems a bit of a reach.

Similarly, in the section of "Hydraulic homeostasis in surviving trees after multi-decadal drought" the authors discuss how the forest experienced both a very pronounced wet and a very pronounced dry season and how the trees maintained their hydraulic function. However, during 2014 and 2015 when there was a lot of drought-induced mortality and loss of biomass (Fig 2a) and productivity (Fig 2c) there were two very dry years back-to-back and the drought stress was likely much greater than what occurred in 2023. If this occurs again with several very dry years in a row there may again be increased drought stress and drought-induced mortality in the experimental plot and a loss of ecohydrological stability. Also, the forest had just experienced a wetter-than-average wet season in 2023 and was likely to be less stressed by a drier-than-average dry season, unlike what occurred in 2014-2015 when there was greater drought for a longer period.

I do not have any concerns regarding flaws in the manuscript including the statistical analysis that should prohibit its publication.

Minor comments:

Line 172: "at our the study site"

Line 391: change "weighted" for weighed

Line 393: change "reweighted" for reweighed

Line 394: change "reweighted" for reweighed

Line 638: same month for the peak of wet (05/2023) and the peak of dry (05/2023) season

Version 1:

Decision Letter:

Dear Pablo,

Thank you for submitting your revised manuscript "The fate of Amazon rain forests under soil drought: collapse or stabilisation?" (NATECOLEVOL-24123431A). On the basis of the revisions and the responses to the reviewers, my colleagues and I have decided that we can proceed with your manuscript without further input from the external reviewers. Therefore, we'll be happy in principle to publish it in Nature Ecology & Evolution, pending some revisions to satisfy editorial points and compliance with journal guidelines.

We will perform detailed checks on your paper and send you a checklist detailing our editorial and formatting requirements once ready. **To do this, we need a copy of the manuscript in an editable format (Microsoft Word or LaTeX), as we cannot proceed with the PDF at this stage. Please email us the file at your earliest convenience to avoid delays.** Do not upload the final materials or make any revisions until you receive our checklist.

[redacted]

Version 2:

Decision Letter:

4th April 2025

Dear Dr Sanchez-Martinez,

We are pleased to inform you that your Article entitled "Amazon rainforest adjusts to long-term experimental drought", has now been accepted for publication in Nature Ecology & Evolution.

Over the next few weeks, your paper will be copyedited to ensure that it conforms to Nature Ecology and Evolution style. Once your paper is typeset, you will receive an email with a link to choose the appropriate publishing options for your paper and our Author Services team will be in touch regarding any additional information that may be required

Due to the importance of these deadlines, we ask you please us know now whether you will be difficult to contact over the next month. If this is the case, we ask you provide us with the contact information (email, phone and fax) of someone who will be able

to check the proofs on your behalf, and who will be available to address any last-minute problems. Once your paper has been scheduled for online publication, the Nature press office will be in touch to confirm the details.

Acceptance of your manuscript is conditional on all authors' agreement with our publication policies (see www.nature.com/authors/policies/index.html). In particular your manuscript must not be published elsewhere and there must be no announcement of the work to any media outlet until the publication date (the day on which it is uploaded onto our web site).

Authors may need to take specific actions to achieve [compliance](https://www.springernature.com/gp/open-research/funding/policy-compliance-faqs) with funder and institutional open access mandates. If your research is supported by a funder that requires immediate open access (e.g. according to [Plan S principles](https://www.springernature.com/gp/open-research/plan-s-compliance)) then you should select the gold OA route, and we will direct you to the compliant route where possible. For authors selecting the subscription publication route, the journal's standard licensing terms will need to be accepted, including [self-archiving and license to publish](https://www.nature.com/nature-portfolio/editorial-policies/self-archiving-and-license-to-publish). Those licensing terms will supersede any other terms that the author or any third party may assert apply to any version of the manuscript.

We welcome the submission of potential cover material (including a short caption of around 40 words) related to your manuscript; suggestions should be sent to Nature Ecology & Evolution as electronic files (the image should be 300 dpi at 210 x 297 mm in either TIFF or JPEG format). Please note that such pictures should be selected more for their aesthetic appeal than for their scientific content, and that colour images work better than black and white or grayscale images. Please do not try to design a cover with the Nature Ecology & Evolution logo etc., and please do not submit composites of images related to your work. I am sure you will understand that we cannot make any promise as to whether any of your suggestions might be selected for the cover of the journal.

You can generate the link yourself when you receive your article DOI by entering it here: <http://authors.springernature.com/share>.

[redacted]

P.S. Click on the following link if you would like to recommend Nature Ecology & Evolution to your librarian <http://www.nature.com/subscriptions/recommend.html#forms>

** Visit the Springer Nature Editorial and Publishing website at http://editorial-jobs.springernature.com?utm_source=ejP_NEcoE_email&utm_medium=ejP_NEcoE_email&utm_campaign=ejp_NEcoE for more information about our career opportunities. If you have any questions please click [here](mailto:editorial.publishing.jobs@springernature.com).**

Comments from the reviewers:

Reviewer #1 (Remarks to the Author):

This study addresses a very consequential theme -- the fate of the Amazon under increased drought. It uses a 20-year throughfall reduction experiment to show that although the drought treatment initially reduced biomass dramatically, the droughted forest reached a new lower-biomass equilibrium with various metrics of hydraulic stress similar to those observed in the control plot. This demonstrates the capacity of forests in this part of the Amazon to equilibrate to dramatic reductions in water availability, suggesting that collapse into a non-forested state is unlikely. This is a valuable insight that advances our understanding of Amazonian forest responses to drought.

Overall, the research methodology is solid, and the paper is well-written, and my concerns are mostly minor.

I do have one general concern: in places, the interpretation goes beyond what I believe to be warranted, in large part because there is limited acknowledgment of the differences between the throughfall exclusion experiment and expected future drought conditions. Importantly, the experiment (1) induced a one-time permanent change, as opposed to the large (and increasing) variability in water availability expected in the future, and (2) induced only soil drought, whereas atmospheric drought (high VPD) is an extremely important component of natural drought and is coupled to soil drought. While there is some good discussion of the latter in the paragraph starting on line 237, the former is not discussed, and both seem a bit under-recognized throughout the manuscript. Specifically, I felt that the title and text on line 109 ("These results rule out drought-

induced biomass collapse...”) were particularly misleading. The language of the entire manuscript should be carefully reviewed to ensure that the interpretation is consistent with the reality that the experiment does not perfectly mimic natural drought.

We thank the reviewer for these thoughtful comments. The reviewer is right in that we did not fully disclose differences between natural drought and the experimental drought in terms of increased climatic variability. We have included some text on this in lines 250-254. After acknowledging the reviewer point on the lack of direct control on climatic variability, we would like to point that the experiment does reflect the climatic variability on the site. Therefore, even if future studies specifically simulating an increase of climatic variability are needed, in this experiment we can indirectly assess how this observed higher variability interact with multidecadal drought. For instance, during 2023, we had a very strong “El Niño”, with the highest VPD ever recorded on the site. These conditions were preceded by a humid wet season, with very low VPD. We expect these extreme conditions, which are likely to be related to climate change, to be captured both in Control and TFE, showing how they interact with sustained long-term soil drought in the TFE plot. We fully agree with the reviewer comment on the language and we have carefully reviewed the manuscript to avoid misleading wording and trying to avoid overgeneralizing. To do so, we included the word “soil drought” in the title as well as made some modifications throughout the manuscript (e.g., Line 110-111, 220, 228).

Specific comments:

Title – Based on the title, I expected the study to give predictions for the Amazon under climate change (increasing drought). While it certainly helps inform our understanding of the Amazon’s future under drought, it falls short of re-creating the future drought conditions in the Amazon.

We believe that our experiment could be seen as a simulation under a potential drier climate, specially in terms of soil drought (which is now explicitly mentioned in the title). This is especially true under “El Niño” years, where there is not only soil drought but also exceptionally high VPD, which are expected to occur together, imposing stressful conditions for Amazonian flora. We believe that looking at the effects of sustained drought plus dry atmospheric conditions happening during the dry season over decades is a valid and logistically suitable approach to understand how Amazonian rain forests will respond to an increasingly dry climate on the field. That being said, we acknowledge that a modification on VPD jointly with soil moisture are needed to fully recreate natural drought, even if logistically challenging (as discussed in lines 245-261). Accordingly, we have included the word soil throughout the text and in the title (lines 2, 220, 228).

Line 45 – define “net wood productivity” is not a conventional term, and most closely approximates (aboveground) woody productivity, a component of net primary productivity that would never be negative. I’d suggest using something like Δ biomass or Δ AGB. (Net ecosystem productivity is also inappropriate because it includes components other than biomass.)

We fully agree with the reviewer and we have changed this wording throughout the text, using annual change in biomass (Δ biomass) instead.

Line 49- can remove “is considered to be”

Done.

Line 67 – reword “poorly unknown”

Done, we used “largely unknown” instead.

Lines 109-113- the language “rule out” goes beyond what can reasonably be inferred from this experiment, as described above.

We agree with the reviewer and we have reworded this sentence. Now it reads as it follows: “These results suggest that drought-induced biomass collapse is unlikely in this ecosystem, indicating that Amazonian rain forests can reach eco-hydrological stability after multi-decadal drought, despite high mortality rates and large reductions in biomass and productivity caused by prolonged severe water deficit.”

Line 124 – biomass relative water availability – (this may be somewhat biased because big trees would have a larger proportion of heartwood)

The reviewer raises a very interesting point here. We agree that separating heartwood from sapwood biomass is an interesting approach to use and we will definitely keep it in mind for future works. However, here we were interested in coupling soil water content with our biomass estimations, which are derived from the trunk diameter and height to calculate its volume and then its mass by means of wood density, as it is the standard procedure. We believe that this metric can be useful in other setups, so we wanted to keep it as simple as possible to allow replication in other places with biomass and soil water content data. Moreover, data on sapwood depth is quite rare in Amazon ecosystems. Therefore, we appreciate the comment as it opens future ideas but we prefer not to include heartwood in the analyses as it brings a degree of complexity that we cannot inform or control in the present.

Line 138 – define “net aboveground productivity”. This is not a conventional term.

This term has been replaced by annual change in biomass (Δ Biomass).

Line 258 – fix grammar

Done, thank you.

Line 295- change 1st sentence to past tense

Done, thank you.

Fig. 1 – Use a different term for “Productivity”, such as Δ Biomass. “Productivity” would most commonly refer to gross or net primary productivity (or one of their components) and would not be negative (except for net ecosystem productivity).

Done, thank you.

Reviewer #2 (Remarks to the Author):

This manuscript is well-written, interesting, and timely. The study suggests that after two decades of drought and substantial drought-induced tree mortality, a one-hectare plot of Amazonian rainforest reached ecohydrological stability. The authors reached this conclusion because the largest trees had died meaning that more water was available for the remaining trees. The authors found that the remaining living trees in the experimental plot had leaf water potential, sap flow, and tissue water content similar to the trees in the control plot. These are very interesting findings with broad implications, especially because recent studies show that the Amazonian forest could reach a tipping point and an ecosystem collapse.

The features of this manuscript that I consider to be of immediate interest are:

- The results that smaller trees (<30 cm diameter) increased their growth during what the authors are calling the stabilization phase is interesting, however, these trees will likely not be able to become very big as the larger trees in the drought experiment died. As the authors describe, these findings are interesting because they shed light on what this forest may look like under future drier climates.
- It is also of immediate interest that once water availability increased after reaching ecohydrological stability productivity also increased, however, it continued to be lower than the control plot. Meaning that the carbon uptake of this forest under a drier climate will be lower than the current levels of carbon uptake.
- During the period of ecohydrological stability there was a strong El Niño episode in 2023 and the forest in the drought experiment remained stable, suggesting that drought-induced biomass collapse may not occurred in this ecosystem under drier conditions. However, this does not necessarily rule this out and please see my comment below.

In lines 109-113, the authors state that their findings rule out drought-induced biomass collapse in this ecosystem; however, this may not be the case if there are more severe droughts like the one in 2014-2015. These results are promising but to say that they “rule out” that this ecosystem will not suffer from biomass collapse due to drought seems a bit of a reach.

We thank the reviewer for their thoughtful comments and suggestions. We agree with the potentially misleading use of the word “rule out”, so we have rewritten the sentence, which now reads as it follows: “These results suggest that drought-induced biomass collapse is unlikely in this ecosystem, indicating that Amazonian rain forests can reach eco-hydrological stability after multi-

decadal drought, despite high mortality rates and large reductions in biomass and productivity caused by prolonged severe water deficit” (lines 110-111).

Similarly, in the section of “Hydraulic homeostasis in surviving trees after multi-decadal drought” the authors discuss how the forest experienced both a very pronounced wet and a very pronounced dry season and how the trees maintained their hydraulic function. However, during 2014 and 2015 when there was a lot of drought-induced mortality and loss of biomass (Fig 2a) and productivity (Fig 2c) there were two very dry years back-to-back and the drought stress was likely much greater than what occurred in 2023. If this occurs again with several very dry years in a row there may again be increased drought stress and drought-induced mortality in the experimental plot and a loss of ecohydrological stability. Also, the forest had just experienced a wetter-than-average wet season in 2023 and was likely to be less stressed by a drier-than-average dry season, unlike what occurred in 2014-2015 when there was greater drought for a longer period.

We agree with the reviewer on the fact that, under even stronger and sustained drought, the forest is likely to keep losing biomass, and we now mention that under a concatenation of multiple droughts, biomass loss and/or the time until stability could be higher (lines 256-257). We also included some discussion about the fact that the exceptional drought in 2023 was preceded by very wet conditions, which could have affected the capability of trees to withstand it (lines 167-172). We argue that we measured similar mean predawn leaf water potentials (as a measure of soil water status) in the leaves at the peak of the dry season compared to a previous study in the same site, with data sampled in October 2016. This points that the soil was similarly dry in both cases. Therefore, we do not expect a very humid wet season to have legacy effects on the maximum stress of trees in this site. Nevertheless, we also disclose that the time at maximum stress may be delayed after an exceptionally humid wet season, and we cannot rule out this possibility.

THE UNIVERSITY
of EDINBURGH

I do not have any concerns regarding flaws in the manuscript including the statistical analysis that should prohibit its publication.

Minor comments:

Line 172: “at our the study site”

Thank you, we eliminated “our” in the sentence.

Line 391: change “weighted” for weighed

Done, thank you.

Line 393: change “reweighted” for reweighed

Done, thank you.

Line 394: change “reweighted” for reweighed

Done, thank you.

Line 638: same month for the peak of wet (05/2023) and the peak of dry (05/2023) season

We corrected this mistake by indicating that the peak of the dry season happened in 10/2023.

Thank you.